# Bone Health in People Living with HIV/AIDS: An Update of Where We Are and Potential Future Strategies

**DOI:** 10.3390/microorganisms11030789

**Published:** 2023-03-19

**Authors:** Musaab Ahmed, Dushyant Mital, Nuha Eljaili Abubaker, Maria Panourgia, Henry Owles, Ioanna Papadaki, Mohamed H. Ahmed

**Affiliations:** 1College of Medicine, Ajman University, Ajman P.O. Box 346, United Arab Emirates; 2Center of Medical and Bio-Allied Health Sciences Research, Ajman University, Ajman P.O. Box 346, United Arab Emirates; 3Department of HIV and Blood Borne Virus, Milton Keynes University Hospital NHS Foundation Trust, Eaglestone, Milton Keynes MK6 5LD, UK; 4Clinical Chemistry Department, College of Medical Laboratory Science, Sudan University of Science and Technology, Khartoum P.O. Box 407, Sudan; 5Department of Geriatric Medicine, Milton Keynes University Hospital NHS Foundation Trust, Eaglestone, Milton Keynes MK6 5LD, UK; 6Department of Rheumatology, Milton Keynes University Hospital NHS Foundation Trust, Eaglestone, Milton Keynes MK6 5LD, UK; 7Department of Medicine and HIV Metabolic Clinic, Milton Keynes University Hospital NHS Foundation Trust, Eaglestone, Milton Keynes MK6 5LD, UK

**Keywords:** HIV, osteoprosis, bisphosphante

## Abstract

The developments in Human Immunodeficiency Virus (HIV) treatment and in the care of people living with HIV (PLWHIV) and Acquired Immunodeficiency Syndrome (AIDS) over the last three decades has led to a significant increase in life expectancy, on par with HIV-negative individuals. Aside from the fact that bone fractures tend to occur 10 years earlier than in HIV-negative individuals, HIV is, per se, an independent risk factor for bone fractures. A few available antiretroviral therapies (ARVs) are also linked with osteoporosis, particularly those involving tenofovir disoproxil fumarate (TDF). HIV and hepatitis C (HCV) coinfection is associated with a greater risk of osteoporosis and fracture than HIV monoinfection. Both the Fracture Risk Assessment Tool (FRAX) and measurement of bone mineral density (BMD) via a DEXA scan are routinely used in the assessment of fracture risk in individuals living with HIV, as bone loss is thought to start between the ages of 40 and 50 years old. The main treatment for established osteoporosis involves bisphosphonates. Supplementation with calcium and vitamin D is part of clinical practice of most HIV centers globally. Further research is needed to assess (i) the cut-off age for assessment of osteoporosis, (ii) the utility of anti-osteoporotic agents in PLWHIV and (iii) how concomitant viral infections and COVID-19 in PLWHIV can increase risk of osteoporosis.

## 1. Background

The metabolic features of PLWHIV are evolving as ARVs developments, efficacies and potencies improve. As PLWHIV age, there is an increasing percentage of patients over the age of 50, reaching more than 50% in European nations and the USA [1,2]. Life expectancy for PLWHIV is now almost equal to that of the HIV-uninfected population. In terms of avoiding and treating comorbidities, PLWHIV are more at risk of developing related conditions than the general population [3,4]. The majority of PLWHIV are currently undergoing sustained uninterrupted ARV treatment and have had undetectable HIV viral loads for long periods of time. The complicated interplay between age, comorbidities and traditional risk factors for bone fragility are particularly prevalent in this population and, to a lesser extent, also affect the well-controlled HIV infection itself. Special considerations are needed for women living with HIV and their bone health. For instance, Ahmed et al. demonstrated that women with HIV are likely to experience early menopause, and this may make them more prone to developing osteoporosis [5,6]. Importantly, HIV also may have negative impacts on the function of the pituitary, thyroid and adrenal glands, and this ultimately may increase the risk of developing osteoporosis [7,8]. Diabetes is also associated with an increase in the risk of fracture [9]. HIV is also associated with a high risk of insulin resistance and type 2 diabetes, dyslipidemia and metabolic syndrome [10]. These metabolic and endocrine factors may directly and indirectly increase the risk of osteoporosis and fracture in PLWHIV. Traditional risk factors include low BMI, age, dietary variables, harmful habits and hypogonadism. Hypogonadism is prevalent among HIV-positive males, affecting roughly 20% of individuals. It might be either primary or secondary hypogonadism associated with obesity, metabolic syndrome, lipodystrophy or dysfunction of the hypothalamus and pituitary axis [11]. Malnutrition and lower frequency of exertive activities have been linked to changes in the bone microstructure in PLWHIV [12]. The combination of a number of risk factors in people co-infected with HCV and HIV might explain the population’s elevated fracture risk. As in the general population, serious falls within the preceding year requiring medical attention are also major predictors of fragility fractures among PLWHIV receiving ARV [13]. Therefore, metabolic and endocrine disorders, especially osteoporosis, are best managed in a dedicated metabolic HIV clinic [14].

## 2. Methods

This narrative review was conducted by examining many databases, including PubMed, Scopus, Cochrane Library, MEDLINE (EBSCO), Web of Science and Google Scholar. The most common search terms used were “HIV AND bone”, “HIV AND osteoporosis”, “HIV AND bisphosphonate” and “Management of osteoporosis AND HIV”. The review was organized into mean headings (prevalence of osteoporosis in HIV population, impact of HIV on the bone, impact of HIV treatment on the bone, endocrine conditions and bone, hepatitis B, hepatitis C and management and prevention of osteoporosis in HIV). The search was based on studies published in English from January 2003 to January 2023. The abstracts and articles were then screened. The total number of publications found in the initial search was 340. The total number of publications selected for the review was 114 (Figure 1). Articles were scanned and read; further relevant references in the reference lists were also included. Information from these articles was summarized in relation to the study design, duration of the study, number of participating patients, medications used and assessment criteria for the safety and effectiveness of medication; then, final conclusions were made. The following criteria were used to select the articles.

### 2.1. Inclusion Criteria

Only randomized controlled trials, clinical trials, reviews and original articles were selected and included in the study. On two occasions, we used case reports, as no further evidence could be found. As a further selection criterion, documents written in English were eligible for this analysis. Based on the topics, all studies concerning HIV, osteoporosis, their prevalence and risk factors and the management of osteoporosis were included.

### 2.2. Exclusion Criteria

Commentaries, letters to the editor, case reports, protocols, news, opinions, theses, notes, short surveys, conference abstracts, repeated studies and papers written in languages other than English were excluded. All the retrieved manuscripts were imported into EndNote software (version 20) to remove duplicates. Then, the titles and abstracts of the studies were screened based on the eligibility criteria by two research team members.

## 3. Prevalence of Fracture in Association with HIV

HIV is a chronic infection that has a direct impact on the bones, and also induces metabolic changes that may lead to negative impacts on the bones. Therefore, it is not surprising that the risk of acquiring a fragility fracture in PLWHIV has been found in meta-analyses to be around 35 to 68% in comparison with the non-HIV infected population [15,16,17,18]. The estimated frequency of acquiring a vertebral fracture in PLWHIV was 22%, while the prevalence was 4.1% to 47% [19]. The variation in fracture risk observed in these studies suggests that different factors are involved in associating HIV with low BMD. For instance, in the meta-analysis conducted by O’Neill et al. and Dong et al., it was shown that co-infection with HCV in PLWHIV is associated with a higher risk of bony fractures than those monoinfected with HIV [12,13]. Fractures are likely to occur in those aged between 40 to 59 years old, and it has been suggested that a fracture is likely to occur 10 years earlier in PLWHIV in comparison with the non-HIV-infected population [20,21,22]. It is suggested that the association of HIV with premature menopause begins at the age of 40 years old in women living with HIV [5]. One intriguing observation was that the risk of a hip fracture in the senior elderly population remained the same for those with and without HIV [23]. This raises the question of whether HIV status can be regarded as an independent risk factor, per se, for a low BMD.

Thomsen et al. showed that HIV status was not independently associated with BMD, and they suggested that traditional risk factors contribute to differences in the prevalence of very low BMD [24]. Different factors have been shown to influence the structure of the bones in PLWHIV and can be associated with increases in both mortality and morbidity. Other factors that lead to increases in mortality and morbidity in PLWHIV and fractures are non-AIDS malignancies, diabetes, dyslipidemia, chronic kidney diseases, lung disease and cardiovascular diseases [25]. Therefore, different factors can lead to a decrease in bone mass in PLWHIV, as stated in Figure 2.

## 4. What Is the Impact of HIV on Bone Mineral Density?

HIV has been shown to affect BMD, microstructure and histomorphometry. Starup-Linde et al. (2020) and Chang et al. (2021) demonstrated that HIV is associated with low BMD [19,26]. HIV has also been shown to affect the microstructure of the bones, especially the trabecular areas in the cortical parts. However, in both young and elderly populations living with HIV, the negative impact of HIV on the cortical parts has been noted [27,28,29]. HIV is also associated with abnormalities in bone formation and mineralization, which occur in the early stages of HIV infection. The treatment with ARV may increase bone remodeling, as demonstrated by increased osteoblast and osteoclast surfaces, but persistent mineralization defects, result in increased osteoid volume [30,31].

The loss of BMD with HIV has been shown to be influenced by various factors.

### 4.1. The BMD before Encountering HIV Infection

The majority of PLWHIV live in sub-Saharan Africa, and this may indicate that these individuals are already exposed to associated risk factors, such as other infections and malnutrition (e.g., low body mass index, low calcium intake). The unfortunate reality is that children in sub-Saharan African living with HIV are known to have low BMD due to malnutrition and other infections. Such children, when treated with ARVs, experience an accelerated process of bone loss. For instance, the use of Tenofovir Disoproxil Fumarate (TDF) was associated with BMD loss during the late stages of adolescence, particularly in females [32].

### 4.2. Loss of BMD Is Likely to Occur at the Age of 50 Years

BMD loss is especially prevalent in women in rural areas in South Africa [22,33,34]. This may be attributed, in part, to the increased risk of premature menopause [5].

### 4.3. Direct Impact of HIV on BMD

In two large meta-analyses, it was shown that HIV is associated with low BMD. Chong et al. and Starup-Linde et al. showed that PLWHIV had lower BMD and greater risks of all fractures, including fragility fractures. Starup-Linde et al. also showed that the risks of fragility fractures and hip fractures were increased. PLWHIV had lower BMD at the hip (Z-score −0.31, 95% CI: −0.46 to −0.27) and lumbar spine (Z-score −0.36, 95% CI: −0.39 to −0.15) compared with the controls. This study also concluded that the reduced BMD did not fully explain the increased fracture risk in PLWHIV [19,26].

### 4.4. Onset of ART Therapy and Bone Density

The impact of ARV on BMD is linked with the current status of the patient’s condition. For instance, HIV is known to decrease BMD, establishing the diagnosis of HIV and initiating ARV therapy in some patients may take up to five years [35]. Therefore, at the time of onset of therapy with ARVs, the BMD is low due to a combination of poor health, weight loss and the direct impact of HIV on the bones. Importantly, the largest decrease in BMD tends to be found one to two years following the initiation of ARV. The START trial clearly established that bone loss with the introduction of ARV is much greater than bone loss caused by HIV infection alone [36]. This transitory acceleration of bone loss is related to the Immune Reconstitution Inflammatory Syndrome (IRIS), and this relates to enhanced bone resorption (through upregulation of the RANKL/OPG pathway) as well as the effects of some medicines on bone metabolism. Between the baseline and 2-year mark, TDF-based ARVs were associated with greater declines in BMD in PLWHIV compared to non-TDF-based ARVs [37,38]. Only 1–2 years after ARV initiation does the degree of bone loss surpass that found during menopause or approach that observed after therapy with glucocorticoids or aromatase inhibitors. With long-term ARV and retroviral activity inhibition, BMD may improve and eventually stabilize. A real assessment at several bone locations showed that BMD was only 3 to 8% lower in HIV-positive males matched for age and BMI [39].

## 5. What Are the Physiological Changes That Lead to Increased Risk of Fracture in PLWHIV?

Research studies have elucidated some of the mechanisms behind the increased bone resorption associated with HIV infection. HIV-transgenic rats showed a drop in BMD, an increase in bone resorption accompanied by normal bone production and a drop in BMI. An upregulation of RANKL expression by B-cells and a concomitant decrease in expression of its physiological modulator, OPG, have been reported [40]. Similarly, in ART-naïve HIV-infected individuals, B-cell RANKL/OPG ratios were observed to correlate with total hip and femoral neck BMD, indicating that B-cell dysregulation accelerates HIV-induced bone loss through an imbalance in the RANKL/OPG ratio [41]. In addition, the initiation of certain ARVs (lopinavir/ritonavir plus TDF/emtricitabine) in these patients provokes an accelerated increase in bone resorption. For instance, the improvement of the CD4 count with ARV was found to be associated with an increase in bone markers such as carboxy-terminal collagen crosslinks (CTX), as well as an increase in RANKL and tumor necrosis factor (TNF) [42]. Importantly, T-cells and B-cells produce the osteoclastogenic cytokines RANKL and Tumour Necrosis Factor (TNF), and this can ultimately decrease bone density [43].

## 6. Impact of HIV on the Bones

HIV modulates the function of inflammatory cells, and this can have a direct impact on bone density. For instance, HIV-infected T-cells can modulate the function of macrophages and lymphocytes and, most importantly, enhance the recruitment of osteoclasts, leading to an increase in RANKL and low expression of osteoprotegerin [41,44]. Unfortunately, additional cellular processes continue to erode the bone; for example, the osteoclast in HIV activates RANKL, tartrate-resistant acidic phosphatase (TRAP) and cathepsin K, and this leads to more osteoclast activity, thus decreasing bone density [40,45,46,47]. Besides decreasing the osteoblasts or osteocalcin, HIV modulates the bone marrow mesenchymal stem cells, and this leads to a release of IL-6, IL-8, alkaline phosphatase, runt-related transcription factor 2 (RUNX-2) and bone morphogenic proteins (BMP-2, BMP-7) [46,47,48]. Therefore, all of these factors work toward decreasing bone formation and increasing osteoclastic activities (bone resorption) [40,45,48].

## 7. Impact of Antiretroviral Treatments on the Bones

Ofotokun et al. showed that ARVs influences bone health irrespective of the indirect and transitory reduction in BMD found following the initiation of any ARV regimen [42,43]. Caution is needed with the introduction of ARV therapy, as the magnitude of changes in bone mass can differ according to the dosage of this medication. Importantly, different studies have clearly shown that TDF is associated with reduced BMD. In a prospective study in naïve HIV-infected adults, the participants were divided into three ARV groups; those on TDF-, tenofovir alafenamide disoproxil (TAF)- and abacavir (ABC)-based regimens were selected. Treatment with TDF is associated with greater bone deterioration at 12 and 48 weeks. TAF seems to present similar early bone deterioration at 12 weeks, which disappears at 48 weeks [49].

Switching from a TDF- to a TAF-based regimen was associated with a decrease in parathyroid hormone [50]. A meta-analysis revealed that PLWHIV on stable ARVs showed a more pronounced decline in BMD when treated with TDF (the annual decrease after the first year was 0.67% in the lumbar spine and 0.35% at the total hip) [19]. Importantly, the switch from a TDF-, emtricitabine- or non-nucleoside reverse transcriptase inhibitor (NNRTI)-based ARV regimen to an ABC, lamivudine or dolutegravir ARV regimen was associated with improvements in BMD in the lumbar spine as well as renal function [51]. TDF was associated with proximal renal tubulopathy and urine phosphate wasting in some patients, both of which seem to be connected to cumulative TDF exposure and may continue even after TDF cessation [52]. Alternatively, it seems that the probability of developing tubulopathy is decreased with TAF [53].

In a prospective open-label, multicenter, randomized trial held in 36 European centers, recruited participants (aged 60 or older) who were virologically suppressed on a TDF-containing regimen were randomly assigned elvitegravir (150 mg), cobicistat (150 mg), emtricitabine (200 mg) or TAF (10 mg) daily, or continued therapy containing TDF (300 mg). At week 48, the mean percentage change in BMD in the spine was 2.24% (SD 3.27) in the elvitegravir-, cobicistat-, emtricitabine- and TAF-based groups, and −0.10% (3.39) in the TDF group (between-group difference, 2.43% [95% CI 1.34–3.52]; *p* < 0.0001). The mean percentage change in hip BMD was 1.33% (2.20) in the elvitegravir, cobicistat, emtricitabine and TAF groups and −0.73% (3.21) in the TDF group (difference, 2.04% [1.17–2.90]; *p* < 0.0001). This study showed that in those aged 60 years or older and those with stable and suppressed viral counts, it is safe to switch from a regimen containing TDF to one containing elvitegravir, cobicistat, emtricitabine or TAF (47) [54]. In virologically suppressed PLWHIV, it was also shown that the switch to dolutegravir with rilpivirine was associated with significant improvement in BMD and bone turnover markers compared with tenofovir TDF [46]. It is worth mentioning that it is important to continue to monitor the risk of fracture in PLWHIV even after stopping TDF, as fracture risk is multifactorial in HIV, as mentioned and highlighted previously. This observation is also endorsed by prior systematic and meta-analysis reviews studying the impact of TDF on BMD in the treatment of HIV, pre-exposure prophylaxis (PrEP) and hepatitis B (HBV). The meta-analysis showed that TDF caused greater decreases in BMD when used for all three indications, and the magnitude of these decreases tended to be greater for HIV treatment compared with PrEP. The risk of fracture was not increased in the PrEP group [55].

## 8. Impact of Hepatitis C on Bones

In two meta-analyses, it was shown that HCV is associated with osteoporosis in PLWHIV. Importantly, HIV/HCV coinfection is associated with a greater risk of osteoporosis and fracture than HIV monoinfection. Therefore, strict measures are needed to guard against osteoporosis in terms of regular screening after the age of 50 years using a DEXA scan [16,17], as the risk of HCV affecting BMD is more pronounced in postmenopausal women [56]. Importantly, Bedimo et al. showed that even before the onset of cirrhosis, chronic HCV infection is an independent risk factor for osteoporosis and fractures among PLWHIV [57]. Carlo et al. concluded that vitamin D deficiency might influence liver disease progression in HIV/HCV-coinfected patients [58]. Another possible mechanism is the interaction between inflammation caused by the two viruses and the immune system, which may lead to upregulation of the receptor activator of the nuclear factor-kappaB ligand (RANKL) and osteoprotegerin (OPG) pathways and, hence, bone resorption, in addition to bone toxicity, as a result of the use of TDF or protease inhibitors to virologically suppress both HBV and HIV infections [59]. It worth mentioning, that eradication of HCV is not associated with improvement in BMD [60].

## 9. HIV and Endocrine/Metabolic Changes Relating to Impact on Bones

HIV infection is known to be associated with a majority of endocrine and metabolic changes. These changes can have a direct impact on the BMD. For instance, hypogonadism is one of the more common endocrine disorders associated with HIV [61]. HIV can also have an impact on the pituitary gland [7]. Therefore, disorders of the hypothalamic–pituitary–adrenal axis resulting in adrenal insufficiency and thyroid dysfunction have also been reported, especially after COVID-19 [8,62]. Other metabolic changes in association with HIV are insulin resistance, type 2 diabetes, non-alcoholic fatty liver disease (NAFLD) and hyperlipidemia (10,14) [63]. Whether these metabolic changes can have an influence on the bone structure in PLWHIV is an issue that remains to be elucidated. However, diabetes, insulin resistance and NAFLD have all been reported to be associated with an increased risk of osteoporosis and fracture in individuals without HIV [64,65,66,67]. Xu et al. studied the association between NAFLD and BMD in 89 HIV-infected patients receiving long-term TDF-based ARV for more than three years. Their data showed that patients with NAFLD showed a worse BMD status than those without NAFLD. The incidence rates of osteopenia (42.86% versus 25.93%) and osteoporosis (17.14% versus 3.70%) were significantly higher in PLWHIV with NAFLD than in those without NAFLD, and the odds ratio (OR) for patients with NAFLD exhibiting a worse BMD status compared with those without NAFLD was 4.49 (95% confidence interval (CI) 1.42, 14.15) [68]. Therefore, further research is needed to establish a link between all of these metabolic changes and the risk of osteoporosis in PLWHIV. This is crucial, as large numbers of PLWHIV are living almost to the same average age as the normal population.

## 10. Prevention and Management of Osteoporosis in PLWHIV

### General Recommendations

All clinicians looking after PLWHIV are expected to be familiar with the following:Encouragement of physical exercise and a balanced diet, with the discontinuation of harmful behaviors where relevant (smoking, excess alcohol intake and obesity);Prevention of falls in the elderly population;Checking vitamin D levels at least once a year and taking supplements of calcium and vitamin D when needed;Education about bone health;Health education for women about the risk of premature menopause and impact on the bones;The need for regular bone screening through measurement of vitamin D and DEXA scans;A DEXA scan is indicated for all postmenopausal women, males older than 50 years, and patients with significant clinical risk factors for fragility fractures, because these patients are more likely to benefit from anti-osteoporotic medications in the case of low BMD [69];The FRAX^®^ tool has been recommended for regular assessment of fracture risk in PLWHIV over the age of 40 in the guidelines [70];A FRAX score should not be recognized as a primary screening tool for bone fragility in PLWHIV, but it may aid in the decision to intervene with anti-osteoporotic medication in the event of significantly reduced BMD (it underestimates the risk of fracture in PLWHIV) [71,72];Consider using ARVs with fewer side effects on BMD, especially with potential renal toxicity linked with bone fragility or renal hypophosphatemia; TDF-sparing regimens using TAF-based treatment or integrase inhibitors can be explored [69,70];The degree of BMD improvement is less in HIV-infected people transitioning from TDF to ABC or integrase inhibitors compared to those receiving single injection of 5 mg zoledronic acid added to TDF [73,74];The addition of bisphosphonates to the cessation of TDF leads to greater increases in BMD than cessation of TDF alone [75];In the event of osteoporosis or a high risk of fracture, it may be important to consider anti-osteoporotic medication even if a bone-friendly ARV regimen has been adopted;PLWHIV receiving TAF or an integrase inhibitor-based regimen saw significantly greater increases in body weight than those receiving TDF [76]. The substitution of TDF with TAF is similarly related with an increase in body weight, obesity and an increase in blood lipid levels [77]. The degree to which changes in fat mass contribute to increased BMD or reduced bone loss reported with TAF or integrase inhibitors is still not yet elucidated.

## 11. Calcium and Vitamin D

HIV is associated with a high prevalence of vitamin D deficiency, which can be up to 80%, even in sun “rich” countries [78,79]. Vitamin D insufficiency is defined by blood 25-OH vitamin D concentrations less than 50 nmol/L (20 ng/mL) [80]. The guidelines of the European AIDS Clinical Society (EACS) recommend testing blood levels of vitamin D in PLWHIV with a positive history of fracture, low BMD, high risk of fracture or any other factors associated with lower vitamin D levels, such as dietary deficiency, dark skin, obesity, malabsorption and chronic kidney disease [81]. In our meta-analysis, it was shown that PLWHIV were prone to having low vitamin D compared with the general population. ARV, older age, lower BMI, lower latitude and male sex may represent risk factors for low vitamin D in PLWHIV [82]. There is considerable research interest in the role of vitamin D as an immunomodulator in HIV, as it decreases inflammation, decreases bone marker turnover and improves the treatment of secondary hyperparathyroidism [83,84]. Furthermore, the decrease in vitamin D plasma levels was also found to be associated with an increase in the production of cytokines [85]. Several studies [86,87] have identified a reduction in parathyroid hormone (PTH) or bone turnover markers. Studies conducted to investigate BMD found indications of a positive effect on BMD in adolescents [88,89]. In ARV individuals treated with calcium (1000 mg/day) and high-dose vitamin D (4000 IU) at the commencement of a efavirenz-/emtricitabine-/TDF-based regimen [90], reduced losses in hip and spine BMD were seen compared to the placebo group. This observation emphasizes the importance of treatment of osteoporosis in PLWHIV and the need to halt the demineralization of the bones [30,31]. It worth mentioning that a beneficial effect of vitamin D supplementation was seen despite the use of efavirenz. Efavirenz has been associated with lower vitamin D levels via modulation of various cytochromes and enzymes involved in the activation or deactivation of vitamin D or vitamin D-binding proteins [91,92,93]. In individuals with weight loss and generally declining health, supplementation should be considered as early as the beginning of HIV infection treatment, as vitamin D and calcium supplements may mitigate the loss of BMD at the onset of ARV. Despite the beneficial and important impact of vitamin D and calcium supplementation, the EACS recommendations indicate maintenance with 800 to 2000 IU of vitamin D per day [81], similar to the recommendation for the normal population. Furthermore, the systematic review and meta-analysis showed that vitamin D (4000 IU/D) and calcium supplementation was related to a significant increase in the spine and hip BMD of participants taking TDF-based drugs [94].

## 12. Anti-Osteoporotic Drugs

### 12.1. Bisphosphonates

Pinzone et al. showed in a meta-analysis that administration of oral and intravenous bisphosphonates was associated with increased BMD at the lumbar spine and total hip over two years in PLWHIV [95]. Bisphosphonates are regarded as first-line treatments in PLWHIV with osteoporosis because clinical evidence suggests they are well-tolerated, safe and have a positive BMD response comparable to that of the general population. After one or two infusions, the effects of zoledronic acid remain for many years, and this was found to extend up to 2–5 years [96,97].

In a clinical trial, zoledronic acid was associated with an improvement in BMD that lasted for 3 years [98]. The protection provided by a single dose of zoledronic acid can last for 48 weeks from the onset of therapy with ARV [99]. Interestingly, Bolland et al. showed that two annual 4 mg doses of zoledronate have positive effects on bone turnover and BMD in men, and these effects persist for at least 11 years after the second dose [100]. A single dose of zoledronic acid protects from BMD loss in PLWHIV without osteoporosis who start ARV [99]. Zoledronic acid offers increased bone protection lasting for 3 years, and is a better alternative to switching away from TDF [73,74]. Huang et al. and Bolland et al. showed that annual zoledronate appears to be a well-tolerated and effective therapy for HIV-associated bone loss [101,102]. Furthermore, zoledronic acid administration in PLWHIV has been shown to be associated with suppression of the expression of RANKL via modulating signaling pathways of miR-101-3p/RANKL, miR-302/PRKACB/RANKL and miR-145/SMAD3/RANKL. It is possible that the ability of zoledronic acid to decrease the expression of RANKL may in part explain the ability of zoledronic acid as a treatment method for osteoporosis in HIV-positive subjects treated with TDF [103], thus providing sufficient evidence showing the beneficial effects of intravenous zoledronic acid. There is no evidence that PLWHIV are at a higher risk for osteonecrosis of the jaw or atypical femoral fractures related to the use of bisphosphonates. The use of intravenous administration may be more favorable, as it also decreases the postulated risk of gut microbial dysbiosis. Alendronate at a dose of 70 mg, in association with vitamin D and calcium, was shown by different studies to be associated with significant improvement in BMD in children and adults. Of these, the study with the shortest duration was 48 weeks [104,105,106,107,108,109,110].

It is worth mentioning that bisphosphonates are considered a first line in the treatment of osteoporosis in PLWHIV. Importantly, in systematic reviews and meta-analyses, even greater improvement in BMD can be achieved in PLWHIV when bisphosphonates are combined with calcium and vitamin D [111]. We have added other medications used in the treatment of osteoporosis to the discussion in this review, as emerging evidence has suggested their potential beneficial effects in the treatment of osteoporosis in PLWHIV.

### 12.2. Denosumab, Teriparatide, Abaloparatide and Romosozumab

There is no robust evidence endorsing the use of these medications in the treatment of osteoporosis in PLWHIV. For instance, to our knowledge, and up to the date of writing this review, there are no published data on the use of abaloparatide and romosozumab as treatments of osteoporosis in PLWHIV. We have included the available evidence regarding denosumab and teriparatide, as there is little research on these formulations. Makras et al. showed in a prospective open-label, 12-month-long multicenter cohort study in male PLWHIV being treated for osteoporosis with ARVs, zoledronic acid and denosumab are efficient and well-tolerated therapeutic options for increasing BMD, at least for the first year of treatment. The major weakness of this trial is that it recruited only 23 PLWHIV (zoledronic acid was given to 10 patients, while denosumab was given to 13 as a control) [112].

In a case report of a 38-year-old woman living with HIV, denosumab treatment given for 4 years was associated with an increase in BMD and a decrease in bone fracture without reactivation of opportunistic infection or HIV infection [113]. Wheeler et al. showed in a case report that Teriparatide is safe and effective for the treatment of osteoporosis in PLWHIV [114]. Therefore, this emphasizes the important role of bisphosphonate as a first line of treatment for osteoporosis in PLWHIV. More clinical trials are needed before denosumab, Teriparatide, abaloparatide and romosozumab can be used in the treatment of osteoporosis in PLWHIV.

## 13. Limitations and Strength of the Study

This review has some limitations. Free search engines were used for the literature search, and we were unable to fully retrieve some articles. We only included studies in the English language, but due to the number of studies included in our review, the limited number of studies which were missed for this reason would likely have a minimal impact on our results. The review was conducted as a narrative, not as a systematic review, and provides up to date information on the prevalence of osteoporosis in the HIV-infected population. We have also provided a comprehensive exploration of the cellular and molecular mechanisms explaining the impact of HIV and HIV treatment on the bones. Another area covered by this review is the presence of common endocrine conditions in HIV, and a summary of the impact of hepatitis B and C on the bones is provided. In the management section, we provided a summary of all the systematic reviews and meta-analyses included in this review. This will allow the readers to have quick update in the field. The design of this narrative review was aimed to allow readers to be quickly updated on the subject, and, at the same time, to have a detailed summary of the topic of HIV and osteoporosis.

## 14. Conclusions

Different factors appear to have impacts on BMD, as summarized in Figure 3. The broad details of HIV’s effects on bone structure are also summarized in Table 1. The evidence shows that HIV is associated with low BMD, and bisphosphonate is the treatment of choice. Management strategies for osteoporosis are provided in Figure 3. Further research is needed to assess (i) the cut-off age for assessment of osteoporosis; (ii) the utility of anti-osteoporotic agents in PLWHIV; (iii) how the concomitant viral infections in PLWHIV can increase the risk of osteoporosis; (iv) whether the increase in the prevalence of diabetes/obesity in PLWHIV may lead to an increase in the prevalence of osteoporosis; (v) the interaction between the gut and the bone, as this is also a promising field for future research; (vi) clinical trials, which are needed to assess the potential benefits of using new or alternative anti-osteoporotic agents; and (V) whether COVID-19 can also have impact on the bones of PLWHIV.

## Figures and Tables

**Figure 1 microorganisms-11-00789-f001:**
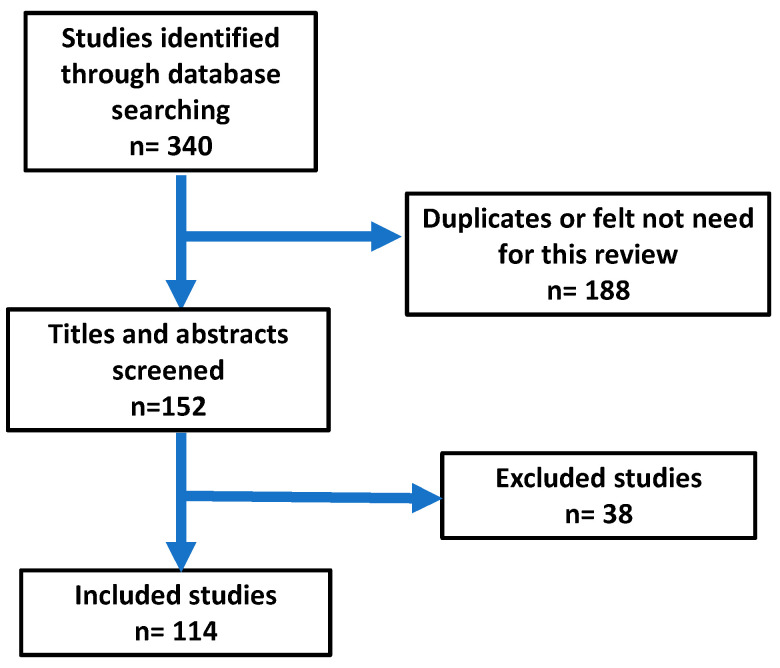
Results of the search of the literature.

**Figure 2 microorganisms-11-00789-f002:**
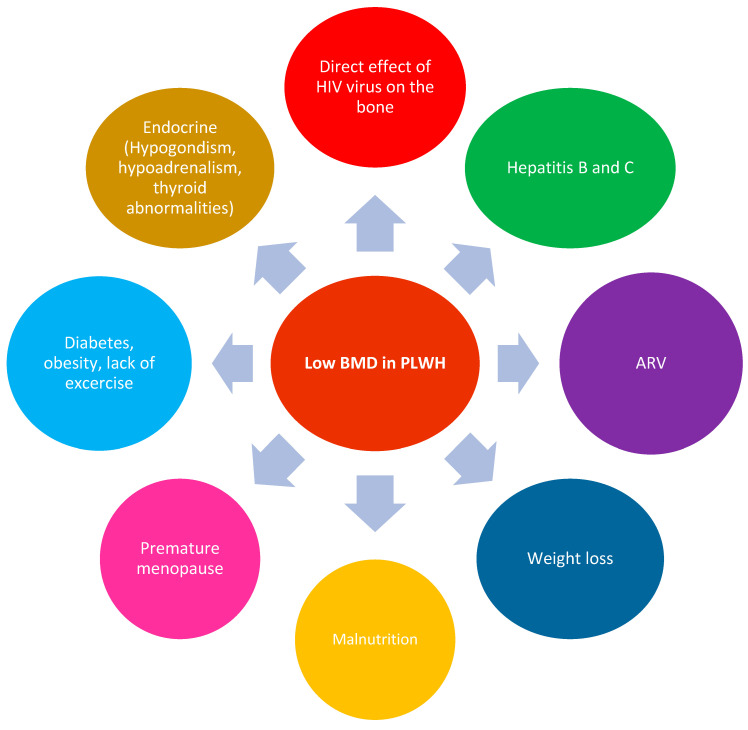
Different factors that may lead to low BMD in PLWHIV.

**Figure 3 microorganisms-11-00789-f003:**
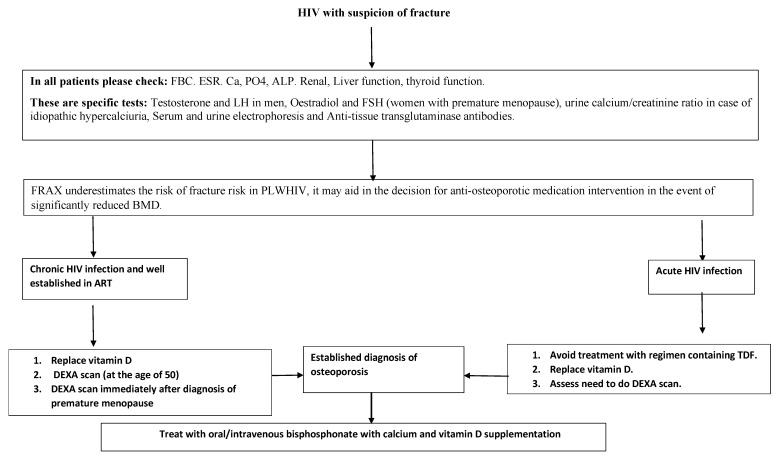
Diagram showing the proposed steps for the management of osteoporosis in acute and chronic HIV.

**Table 1 microorganisms-11-00789-t001:** Summary of all the systematic reviews and meta-analyses included in this review. This will allow the readers to have detailed summary of updates in the field of HIV. TDF (tenofovir disoproxil fumarate); BMD (bone mineral density).

Theme	Main Outcome of Systematic Review or Meta-Analysis	References
Prevalence of fracture in PLWHIV	Fragility fracture in HIV was found in meta-analyses to be around 35 to 68% in comparison with the non-HIV-infected population.	[15,16,17,18]
Frequency and prevalence of vertebral fracture	The estimated frequency of vertebral fracture in PLWHIV was 22%, while the prevalence was 4.1 to 47%.	[19]
HIV and BMD	PLWHIV had lower bone mineral density and increased fracture risk.	[26]
HIV, ARV and BMD	The prevalence of osteopenia/osteoporosis in HIV-infected and antiretroviral therapy (ARV)-treated individuals was increased by two times compared to controls.	[33]
HIV, osteoporosis and viral hepatitis	Hepatitis C and B were associated with osteoporosis in PLWHIV	[16,17]
HIV, TDF and BMD	The impact of TDF in BMD in treatment of HIV, pre-exposure prophylaxis (PrEP) and HBV was assessed. The meta-analysis showed that TDF caused greater decreases in BMD when used for all three indications, and the magnitude of this decrease was greater for HIV treatment compared with PrEP. The risk of fracture did not increase in the PrEP group.	[55]
Diabetes and osteoporosis	A high prevalence of osteoporosis was observed among individuals with type 2 diabetes around the globe.	[65]
HIV and Vitamin D	PLWHIV were prone to having low Vitamin D compared with the general population, and risk factors were ART, older age, lower BMI, lower latitude and male sex.	[94]
Impact of Vitamin D supplements on BMD in PLWHIV	Vitamin D (4000 IU/D) and calcium supplementation was related to a significant increase in BMD in the spine and hip of participants taking TDF-based drugs.	[106]
Bisphosphonates and osteoporosis in PLWHIV	Oral and intravenous bisphosphonates were associated with increased bone mineral density at the lumbar spine and total hip over two years in HIV-positive patients.	[107]
The combination of bisphosphonate, Vitamin D and calcium	Better improvement in BMD was achieved in PLWHIV when bisphosphonates were combined with calcium and vitamin D	[111]

## Data Availability

This review article and information collected from published manuscripts.

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
