# Peer review of "Bone Health in People Living with HIV/AIDS: An Update of Where We Are and Potential Future Strategies"

_microorganisms, 2023, doi:10.3390/microorganisms11030789_

Round 1

Reviewer 1 Report

Authors should seek external reviewer/editor with good English writing skills to help edit the manuscript.

Overall good review of the literature for reports of BMD, and overall bone health in PLWH compared to non-HIV infected counterparts.

Discussion of COVID infection and microbiotim was bit off target for focus on HIV and bone density risk.

Author Response

We thank the reviewer for the comment and opportunity to revise the manuscript:

  1. the paper is revised again by the authors Dr Mital and Dr Henry Owles both are native English speaker and the changes can be seen in red inside the manuscript
  2.  We have deleted the section about COVID-19 and the bone

Reviewer 2 Report

The article is coherent and addresses an important issue about subjects living with HIV/AIDS. In addition, it is a very extensive document that addresses many different aspects of bone health in HIV/AIDS patients. 

The information is helpful and can be applied. However, there are some aspects that the authors should adjust before considering publication.  

It is not specified if the review is systematic or not and if it was designed as a systematic review because some research and information retrieval strategy (PICO, for example) was not used. The authors should clarify this issue in the Discussion. In addition, using only six keywords,  HIV, Osteoporosis, Prevalence, Risk factors, 67 COVID-19, and management, makes the information very varied; if this was the objective contemplated in the research design, it should be mentioned. 

In such a way, dealing with a topic as broad as bone health, although important and necessary, implies the recovery of a lot of information, its classification, and the identification of valuable results. Therefore, it would be very useful if the authors could describe the parameters they used to organize the information. 

In particular, the manuscript has many typographical mistakes, e.g., PLWH or PLWA/H or PLHIV; PLWIH.    

Chapter 11, "Impact of COVID-19 on bone," could be unnecessary because no specific studies exist. Therefore, the authors could consider eliminating it. 

Subheading 6 is not relevant; consider if they remove it. 

Author Response

We thank the reviewer for the comment and opportunity to revise the manuscript:

  1. the paper is revised again by the authors Dr Mital and Dr Henry Owles both are native English speaker and the changes can be seen in red inside the manuscript
  2.  We have deleted the section about COVID-19 and the bone as well as subheading 6
  3. We have mentioned in the method section this narrative review and the limitation we also mentioned this was narrative review and not systematic review 
  4. Please see all the changes inside the manuscript uploaded 

Round 2

Reviewer 2 Report

I insist that the authors should clarify in the material and method what was the design of their search strategy and what considerations they took into account to organize such extensive information, and in the discussion why they designed their search strategy in that way.    

Author Response

please see our response in two sections (A) methods section (B) discussion section under the limitation and strength of the study (whole updated version of manuscript with changes in red enclosed as an attachment) 

(A) 

. Methods

This narrative review was conducted by examining many databases like: PubMed, Scopus, Cochrane Library, MEDLINE (EBSCO), Web of Science and Google Scholar.  The The most common search terms used were “HIV AND bone”, “HIV AND osteoporosis”, “HIV AND bisphosphonate” “Management of osteoporosis AND HIV”. The review was organised into mean headings (prevalence of osteoporosis in HIV population, impact of HIV on the bone, impact of HIV treatment on the bone, endocrine conditions and bone, hepatitis B, C and management and prevention of osteoporosis in HIV). The search is based on studies published in English from January 2003 to January 2023. The abstracts and the articles were then screened. The total number of publications found in the initial search was 340. The total number of publications selected for the review was 114 (figure 1). Articles were scanned and read; further relevant references in the reference lists are also included. Information from these articles was summarised in relation to the study design, duration of study, number of participating patients, medications used, assessment criteria for medication safety and effectiveness, and final conclusions were made. The following criteria were used to select the articles:

2.1. Inclusion criteria:

Only randomized controlled trials, Clinical trials and reviews, original articles were selected and included in the study. On two  occasions, we used Case Reports as no further evidence can be found.  As a further selection criterion, documents in written English were eligible for this analysis. Based on the topics, all studies about HIV and Osteoporosis, Prevalence and Risk factors and Management of Osteoporosis were included

2.2. Exclusion criteria:

Commentaries, letters to the editor, case reports, protocols, news, opinions, theses, notes, short surveys, conference abstracts, repeated studies, and papers written in other languages than English were excluded. All the retrieved manuscripts were imported into EndNote software to remove the duplicates. Then, the titles and abstracts of the studies were screened based on the eligibility criteria by two research team members.

Limitations and strength of the Study

This review has some limitations. Free search engines were used for literature search and the inability to fully retrieve some articles. We only included English-language Studies, but due to the number of studies included in our review, a limited number of studies missed for this reason would likely have minimal impact on our results. The review was conducted as narrative and not as systematic review. The review provided up to date information on the prevalence of osteoporosis in HIV population. We have also provided comprehensive exploration of the cellular and molecular mechanisms explaining the impact of HIV on the bone and impact of HIV treatment on the bone. Another area covered by the review is common endocrine conditions in HIV and summary of the impact of hepatitis B, C on the bone. In the management section, we provided summary of all the systematic review and meta-analysis included in this review. This may allow the readers to have quick update in the field. The design of this narrative review was aimed to allow readers to have quick update on the subject and at the same time to have detailed narration on the topic of HIV and osteoporosis.

Round 3

Reviewer 2 Report

The authors have attended to all suggestions and comments. Therefore, in my opinion, the paper is "Accepted to publish."